# Association Mapping of Amylose Content in Maize RIL Population Using SSR and SNP Markers

**DOI:** 10.3390/plants12020239

**Published:** 2023-01-04

**Authors:** Kyu Jin Sa, Hyeon Park, So Jung Jang, Ju Kyong Lee

**Affiliations:** 1Department of Applied Plant Sciences, College of Agriculture and Life Sciences, Kangwon National University, Chuncheon 24341, Republic of Korea; 2Interdisciplinary Program in Smart Agriculture, Kangwon National University, Chuncheon 24341, Republic of Korea

**Keywords:** molecular markers, normal and waxy maize, population structure, QTLs, UPGMA dendrogram, *Zea mays*

## Abstract

The ratio of amylose to amylopectin in maize kernel starch is important for the appearance, structure, and quality of food products and processing. This study aimed to identify quantitative trait loci (QTLs) controlling amylose content in maize through association mapping with simple sequence repeat (SSR) and single-nucleotide polymorphism (SNP) markers. The average value of amylose content for an 80-recombinant-inbred-line (RIL) population was 8.8 ± 0.7%, ranging from 2.1 to 15.9%. We used two different analyses—Q + K and PCA + K mixed linear models (MLMs)—and found 38 (35 SNP and 3 SSR) and 32 (29 SNP and 3 SSR) marker–trait associations (MTAs) associated with amylose content. A total of 34 (31 SNP and 3 SSR) and 28 (25 SNP and 3 SSR) MTAs were confirmed in the Q + K and PCA + K MLMs, respectively. This study detected some candidate genes for amylose content, such as GRMZM2G118690-encoding BBR/BPC transcription factor, which is used for the control of seed development and is associated with the amylose content of rice. GRMZM5G830776-encoding SNARE-interacting protein (KEULE) and the uncharacterized marker PUT-163a-18172151-1376 were significant with higher R2 value in two difference methods. GRMZM2G092296 were also significantly associated with amylose content in this study. This study focused on amylose content using a RIL population derived from dent and waxy inbred lines using molecular markers. Future studies would be of benefit for investigating the physical linkage between starch synthesis genes using SNP and SSR markers, which would help to build a more detailed genetic map and provide new insights into gene regulation of agriculturally important traits.

## 1. Introduction

Maize (*Zea mays* L.) is the most important agricultural crop. Starch is the main component of maize kernels and comprises about 70% of kernel dry weight. Maize starch is a major source of human food, animal feed, and industrial materials, including for bioethanol production, paper, textiles, etc. [1,2]. Enhancing maize starch content can provide higher kernel quantity and quality, and it can also improve its value in industrial applications [3]. Therefore, manipulating starch content and texture in the kernel is an important target in maize breeding [4]. Maize starch can be differentiated into two types based on the texture and content of the endosperm in the kernel: amylose of the linear form and amylopectin of the branched form. The ratio of amylose to amylopectin in maize kernels is important for the appearance, structure, and quality of food products and processing, such as starch gelling, the firmness and formation of crystalline granules, and the thickening of paste [5,6]. The amylose content in particular exhibits various variations in the natural population of maize, ranging from 0% (waxy maize) to 64% (amylomaize) [7]. Based on the composition of kernel starch, maize can be differentiated into two types: normal (nonwaxy) and waxy maize. In the world in general, normal maize is more cultivated in terms of production amount and area and is mostly used for food and feed. On the other hand, waxy maize, known as sticky or glutinous maize, is a unique type of cultivated maize, the immature ears of which are mainly used for food in East and South-East Asian countries [8,9]. Starch of the normal maize endosperm is made up of approximately 25% amylose and 75% amylopectin [1], whereas starch of the waxy maize endosperm consists of over 99% amylopectin.

Many enzymes and genes play critical roles in maize starch synthesis, such as sucrose synthase (SUS encoded by *shrunken1* (*sh1*); cleavage of sucrose into fructose and UDP-glucose), ADP-glucose pyrophosphorylase (AGPase; small and large units encoded by *brittle2* (*bt2*) and *shrunken2* (*sh2*), respectively), soluble starch synthases (SSs), granule-bound starch synthase (GBSS; encoded by *waxy1* (*wx1*)), starch-branching enzyme (BE; encoded by *amylose extender1* (*ae1*)), and starch-debranching enzyme (DBE; encoded by *sugary1* (*su1*)) [10]. Among such genes, the *Waxy1* (*Wx1*) gene, 3.93 kb long and containing 14 exons on the long arm of chromosome 9, encodes GBSS, which controls amylose synthesis in maize endosperm [11,12]. Although the *Wx1* (dominant wild-type) gene can convert ADP-glucose to amylose, the *wx1* (recessive mutant) gene vastly inhibits the conversion to amylose, which leads to increased accumulation of amylopectin [13]. Homozygous recessive *wx* alleles result in the waxy maize endosperm having a sticky texture and comprising almost only amylopectin [14]. Moreover, the recessive *wx1* gene is also related to the tasty and savory flavor in maize kernels [15]. The homozygous recessive *wx* mutant (waxy maize) was first discovered in the early 1900s in China, and it is widespread in other Asian countries [16]. There are more than 50 natural mutations in the *wx* gene that have been identified at the molecular level [17].

In plant-breeding programs, determining the genetic basis of agronomic traits is a very important scientific issue for crop improvement [18]. Traditional breeding depends on phenotypic selection in the field, which is laborious and time-consuming [19]. However, marker-assisted selection (MAS) enables the selection of desired phenotypes based on genotypes [20]. Discovery of quantitative trait loci (QTLs) is the first step for MAS [21]. Two methods are usually utilized to identify QTLs or genes associated with important agronomic traits: one is QTL mapping based on linkage within segregating populations, which generates crosses between biparents with contrasting phenotypes and genotypes [22]; while the other is association mapping using linkage disequilibrium (LD) between markers and agronomic traits of interest [23,24]. QTL mapping is a classical method for detecting QTLs or genes for quantitative traits of interest without prior genetic knowledge. Many QTLs for starch content in maize have been reported in different biparental populations [25,26,27,28,29,30,31,32,33]. However, low resolution is a critical disadvantage for QTL mapping, which often ranges from 10 to 30 cM [34]. The best way to improve the resolution is to increase the marker density for QTL mapping [35]. With advances in genomics and genotyping technologies, single-nucleotide polymorphism (SNP) markers have been applied to increase marker density with high throughput and little time and cost [4]. Moreover, association mapping using SNP markers is an alternative to traditional QTL mapping using markers such as simple sequence repeats (SSRs) or microsatellites, amplified fragment length polymorphisms (AFLPs), and restriction fragment length polymorphisms (RFLPs). Compared with traditional QTL mapping approaches, association mapping using SNP markers enables the detection of 10 s of 1000 s of loci simultaneously and allows for higher mapping resolution [23,36].

Therefore, the main objectives of this study were to identify QTLs controlling amylose content in maize through association mapping with SSR and SNP markers and to identify candidate markers responsible for amylose content for marker-assisted selection (MAS). In other words, this study tried to detect unknown minor genes that regulate the amylose content of maize kernels.

## 2. Results

### 2.1. Phenotypic Analysis for Amylose Content in RIL Population

Phenotypic variations for amylose content in two parental lines and an 80-recombinant-inbred-line (RIL) population are shown in Figure 1 and Table 1. Amylose content in Mo17 (female) and KW7 (male) was 14.4 ± 0.6 and 2.9 ± 0.6%, respectively. Moreover, the average value of amylose content for the 80 RIL population was 8.8 ± 0.7%, ranging from 2.1 to 15.9%. Based on the average value (8.8%) for the RIL population, we divided the population into two different groups: a low-amylose group and a high-amylose group. The low group consisted of 36 inbred lines, which had an average value of amylose content of 3.4 ± 0.5%, ranging from 2.1 to 4.7%. Meanwhile, the high group contained 44 inbred lines, which had an average value of 13.1 ± 0.9%, ranging from 10.3 to 15.9 (Figure 1, Table 1). To confirm differences in amylose content variation between the low and high groups, a *t*-test was performed using the phenotypic data (Table 1), and this showed statistically significant differences between the low group and the high group (*t* = −45.403, *p* < 0.001).

### 2.2. Cluster Analysis and Population Structure

An unweighted pair group method with arithmetic mean (UPGMA) dendrogram was determined to detect a clustering pattern for the 80 RIL population with 14,968 SSR and SNP markers, and the population was clustered into three major groups (Figure 2a). Group I consisted of a total of 27 lines with 17 low- and 10 high-amylose lines; Group II included 30 lines, which consisted of 9 low- and 21 high-amylose lines; and Group III only included 23 lines with 10 low- and 13 high-amylose lines (Figure 2a). In addition, to understand the population structure among the 80 RIL population, we used a model-based approach via STRUCTURE software using the 14,968 SSR and SNP markers. Although some lines belonged to an admixed group, almost all the lines were divided into two major groups at K = 2 (Figure 3a). Based on a membership probability threshold of 0.8, all lines were divided into three groups: Group I, Group II, and the admixed group. In detail, Group I included 28 lines with 10 high- and 18 low-amylose lines; Group II included 16 lines consisting of 12 high- and 4 low-amylose lines; and the admixed group included 36 lines, which consisted of 22 high- and 14 low-amylose lines (Figure 3a).

### 2.3. Marker–Trait Association for Amylose Content

This study detected loci associated with amylose content using two different mixed linear models (MLMs): a population structure (Q)+ kinship (K) matrix model (Q + K MLM) and a principal component analysis (PCA) + K model (PCA + K MLM). Based on a false discovery rate (FDR) at 0.05 and 0.01, this study found 53 (44 SNP and 9 SSR) and 38 (35 SNP and 3 SSR) marker–trait associations (MTAs), respectively, associated with amylose content using Q + K MLM (Table 2). On the other hand, 40 (37 SNP and 3 SSR) and 32 (29 SNP and 3 SSR) MTAs were detected using PCA + K MLM (Table 3). Moreover, based on Bonferroni thresholds at α = 0.05 and α = 0.01, some MTAs were excluded from those based on the FDR in both Q + K and PCA + K MLM. From this result, a total of 34 (31 SNP and 3 SSR) and 28 (25 SNP and 3 SSR) MTAs, respectively, were confirmed in the two different analyses (Table 2 and Table 3). The percentage of phenotypic variation explained by R^2^ ranged from 20.6% to 87.0% in Q + K MLM for each SNP and SSR marker (Table 2). In the case of PCA + K MLM, phenotypic variation for each marker ranged from 16.6% to 61.9% (Table 3). All SNP and SSR markers detected in PCA + K MLM were overlapped with markers identified by Q + K MLM. Moreover, all significant MTAs for amylose content were only detected on chromosome 9 in both of the MLM analyses and with the two different thresholds.

## 3. Discussion

To understand the genetic relationships and population structure of the 80 RIL maize population, we utilized two methods: a model-based approach and a distance-based approach (Figure 2 and Figure 3). The model-based STRUCTURE revealed that the RIL population could be divided into two major groups and an admixed group at K = 2 (Figure 2), while the distance-based UPGMA dendrogram showed the RIL population divided into three major groups with 47.8% of genetic similarity (Figure 3). Although the 80 RIL maize population was well divided into two or three major subgroups using 14,968 markers, there was no clear group pattern in accordance with amylose content (Figure 2a and Figure 3a). The admixed group contained 45% of the total lines and was a mixture of high- or low-amylose lines. Furthermore, to better comprehend the genetic relationships and population structure of the maize 80 RIL population, we analyzed the model-based and distance-based approaches of the RIL population using 40 overlapping SNP and SSR markers (Table 3) related to amylose content by association mapping. In the model-based approach, the two major groups were Group I, which included 27 low-amylose inbred lines and only one high-amylose line (RIL67), and Group II, which comprised 41 inbred lines consisting of 39 high-amylose and two low-amylose lines (RIL12 and RIL66) (Figure 2b). Moreover, in the distance-based approach the maize 80 RIL population was divided into two major groups with 10.1% genetic similarity: Group I included 33 low-amylose and 2 high-amylose lines (RIL14 and RIL67), and Group II contained 42 high-amylose and 3 low-amylose lines (Figure 3b). As shown by these results, the 40 SNP and SSR markers clearly distinguished high and low amylose content in the maize 80 RIL population.

Association mapping is a powerful method to identify loci and genes related to important agronomic traits and uses high-throughput genotyping technologies in crop [37]. Population structure in MTAs is a critical factor because it can increase the probability of false-positive associations [38]. To resolve this problem, several models have been developed for association mapping, such as the Q + K model and PCA model [39,40]. Although much research has generated a Q-matrix for association mapping by STRUCTURE software, this method is computationally intensive and not suitable for running very large datasets [37]. The PCA model is also applied to generate a population structure and is known to have fewer residual false-positive associations [41]. Therefore, this study used the Q + K and PCA + K models to identify MTAs for amylose content and to minimize false-positive associations. In the results, 13 more MTAs (7 SNPs and 6 SSRs) for amylose content were detected with Q + K MLM than with PCA + K MLM (Table 2 and Table 3). However, 40 common MTAs were confirmed for Q + K MLM and PCA + K MLM. To avoid false positives and false negatives, the correct *p*-value threshold should be determined for statistical significance. Many statistical procedures accounting for multiple testing have been proposed to select statistical significance thresholds in association mapping. The Bonferroni correction and FDR are commonly used for crops [42]. However, the Bonferroni correction method is considered the most conservative, while the FDR method is a popular, less conservative approach for selecting MTAs [36]. These methods were used to detect significant MTAs in this study. With the FDR method, a total of 53 and 40 MTAs were confirmed in Q + K MLM and PCA + K MLM, respectively. When the more conservative Bonferroni correction was used, we found 34 and 28 MTAs in Q + K MLM and PCA + K MLM, respectively (Table 2 and Table 3). In particular, umc1634 at GRMZM2G121333 was the most statistically significant marker in association mapping at *p* = 2.64 × 10^−11^ and 1.74 × 10^−10^ with R^2^ of 0.870 and 0.619 in Q + K MLM and PCA + K MLM, respectively.

To find candidate genes for amylose content, we confirmed the genomic region of the 40 overlapping MTAs between Q + K MLM and PCA + K MLM. Among these MTAs, 8 MTAs were unknown, and 32 SNP and SSR marker associations with 26 genes were detected in the Bin 9.02~9.03 region, which corresponds to a physical position between 20,346,749 to 100,030,452 base pairs (bps). Moreover, in our previous studies [43,44] for QTL mapping, we detected two QTL regions between umc1634 and *wx1* using SSR and SNP markers and between PZE-109022525 and PZE-109024175 using SNP markers. The 14 statistically significant markers at 9 genes (GRMZM2G147319, GRMZM2G024993, GRMZM2G121333, GRMZM2G396553, GRMZM2G118355, GRMZM2G118687, GRMZM2G118690, GRMZM2G082855, GRMZM2G370155) in the Q + K MLM and PCA + K MLM were detected between previous QTL intervals (Table 2 and Table 3). Although many genes were detected in the association mapping using SNP markers, *wx1* at GRMZM2G024993 was only associated with amylose content and encodes GBSS for amylose biosynthesis. This finding is not surprising because this gene on chromosome 9 was highly linked and well known for its association with amylose content [5]. However, one candidate gene is GRMZM2G118690-encoding BBR/BPC transcription factor, which is for control of seed development [45]. This transcription factor is also greatly associated with the amylose content of rice [46]. Moreover, some significant genes were identified with highly phenotypic variation (R^2^). For example, umc2213 at GRMZM5G830776-encoding SNARE-interacting protein (KEULE) had *p*-values of 3.66 × 10^−10^ and 3.18 × 10^−9^ with R^2^ of 0.698 and 0.481 in Q + K MLM and PCA + K MLM, respectively (Table 2 and Table 3). SYN10618 and SYN10617 at GRMZM2G092296 were also significantly associated with amylose content in this study. Although the PUT-163a-18172151-1376 marker is uncharacterized, this marker also statistically significantly associated with amylose content with high phenotypic variance in this population. Although we found only the *wx1* gene as a key gene in the starch biosynthesis pathway, a previous study identified potential candidate genes that encode enzymes in nonstarch metabolism and act as regulators of starch biosynthesis [4].

Starch content as a quantitative trait is known to be controlled by a large number of genes/QTLs that each have small effects [47]. This study found some genes on chromosome 9 associated with amylose content. Many studies already found QTL or genes associated with starch and amylose content in chromosome 9 in maize [4,5,27,32,37,48]. These QTLs were detected in the intervals of bnlg430-dupssr19 and umc1771-bnlg430 on chromosome 9 [48] and in the interval of PZE10907827-PZE109082140 with the physical position of 126.2–130.8 Mb [4] and at 24,028,939 bp and 126,584,775 bp on chromosome 9 [49]. From genome-wide association studies (GWAS), several SNPs on chromosome 9 showed significant association with amylose content [5]. Our findings were inconsistent with those of other studies for most genetic regions or genes, although chr.9.S_23283117 at GRMZM2G024993 [5] was identified in this study. This difference in the results for chromosome 9 could be because of genetic background, population size, recombination events, environmental effects, and analysis methods [50].

This study only focused on amylose content using an RIL population derived from dent and waxy inbred lines. Therefore, this study cannot identify other genes or QTLs related to the starch biosynthesis pathway. Future studies would be of benefit for investigating the physical linkage between starch synthesis genes using SNP and SSR markers, which would help to build a more detailed genetic map and provide new insights into gene regulation of agriculturally important traits. Moreover, our future work will focus on designing SNP markers and QTLs related to the starch biosynthesis pathway in maize-breeding programs.

## 4. Materials and Methods

### 4.1. Plant Material and Amylose Content Evaluation

A population of 80 F_7:8_ RILs used in this study was constructed by single-seed descent between normal maize inbred line Mo17 (United States Corn Belt) and waxy maize inbred line KW7 (Korean waxy maize landrace) at the Maize Experiment Station, Gangwon Agricultural Research and Extension Service, Hongcheon [43,51]. Analysis of amylose content was performed on both parental lines as well as on the 80 RILs in kernel, and this study used three replicates for each line and an average value [44].

### 4.2. SSR and SNP Genotype in RILs

A total of 14,968 SSR and SNP genotypes for RILs were generated in this study from two different studies for genetic and QTL mapping [43,44]. Among them, the 546 informative markers (541 SSRs and 5 SNPs) reported in a previous study by Sa et al. (2015) [43] were obtained after filtering by a chi-square test (*p* < 0.05) [43], and the 14,422 SNP markers reported in a previous study by Sa et al. (2021) [44] were obtained for association mapping after filtering of the Illumina MaizeSNP50 Bead-Chip (Illumina, Inc. San Diego, CA, USA) of 56,110 maize SNPs, which covers 19,540 genes developed from the B73 reference sequence [44,52]. Among the 56,110 SNPs, many SNPs were removed, such as unanchored SNPs; missing, heterozygote, and monomorphic markers in parental lines; over 10% missing in the RIL population; and SNPs on duplicated positions. Moreover, we also excluded skewed SNP markers by a chi-square test (*p* < 0.05).

### 4.3. Statistical Data Analysis

Genetic similarity (GS) was generated for each pair of lines of the RIL population using the Dice similarity index [53]. The similarity matrix was used to construct a UPGMA dendrogram with the application of SAHN clustering in NTSYS-pc V2.1 [54]. This study used the model-based program STRUCTURE 2.3 [55] to confirm the population structure for the 80 RIL population. The membership coefficient of each K value at each subpopulation was obtained with 5 replicates ranging from 1 to 10 using 100,000 cycles for both burn-in and run length. The delta K statistic suggested by Evanno et al. (2005) [56] was calculated with STRUCTURE HARVESTER (http://taylor0.biology.ucla.edu/structharvest/, accessed on 1 April 2022) based on the STRUCTURE results. Each RIL with membership probabilities <0.80 was assigned to a mixed group. Association analysis was performed using TASSEL 3.0 software [57], which was used to evaluate MTAs using an MLM (Q + K MLM and PCA + K MLM) with Q, PCA, and K matrix at a significance value of *p* ≤ 0.05. The PCA and K matrix were calculated using the PCA and kinship option, respectively, in TASSEL 3.0 software from marker data. The statistical significances of the SSRs and SNPs were evaluated with FDR-adjusted *p*-values at 0.05 and 0.01 critical thresholds [58] and Bonferroni-corrected thresholds were performed to control the type I error rate at both α = 0.05 and α = 0.01 as the cut-offs (3.34 × 10^−6^ and 6.68 × 10^−7^, corresponding to “−log10p (α/*n*)” values of 5.48 and 6.18, respectively).

Basic statistics used Microsoft Office Excel (2016) (Microsoft Corporation) for average, standard deviation, and minimum and maximum values in parental lines and RILs. The *t*-test for difference between the low and high groups for amylose content was calculated using IBM SPSS Statistics version 26 (IBM Corporation, Armonk, NY, USA).

## 5. Conclusions

Maize can be differentiated based on the composition of kernel starch into two types: normal maize and waxy maize. Starch of the normal maize endosperm is approximately 25% amylose and 75% amylopectin, whereas starch of the waxy maize endosperm consists of over 99% amylopectin. This study aimed to identify QTLs controlling amylose content in maize through association mapping with SSR and SNP markers and to identify candidate markers responsible for amylose content for MAS. Amylose content in Mo17 (female) and KW7 (male) was 14.4 ± 0.6 and 2.9 ± 0.6%, respectively. Moreover, the average value of amylose content for the 80 RIL population was 8.8 ± 0.7%, ranging from 2.1 to 15.9%. Based on the average value (8.8%) for the RIL population, we divided the population into two different groups: a low-amylose group and a high-amylose group. A dendrogram using UPGMA was used to detect clustering patterns for the 80 RIL population with 14,968 SSR and SNP markers, and the population was found to be clustered into three major groups. We used two different analyses—Q + K and PCA + K MLMs—and all the SNP and SSR markers detected in PCA + K MLM were overlapped with markers identified by Q + K MLM. Moreover, GRMZM2G118690-encoding BBR/BPC transcription factor controls seed development and is associated with the amylose content of rice. The umc2213 at GRMZM5G830776-encoding SNARE-interacting protein (KEULE) was significant, with a higher R^2^ value in two different methods. SYN10618 and SYN10617 at GRMZM2G092296 were also significantly associated with amylose content. The uncharacterized marker PUT-163a-18172151-1376 is also statistically significantly associated with amylose content with high phenotypic variance. This study focused on amylose content using an RIL population derived from dent and waxy inbred lines. Future studies would be of benefit for investigating the physical linkage between starch synthesis genes using SNP and SSR markers, which would help to build a more detailed genetic map and provide new insights into gene regulation of agriculturally important traits in maize molecular breeding studies.

## Figures and Tables

**Figure 1 plants-12-00239-f001:**
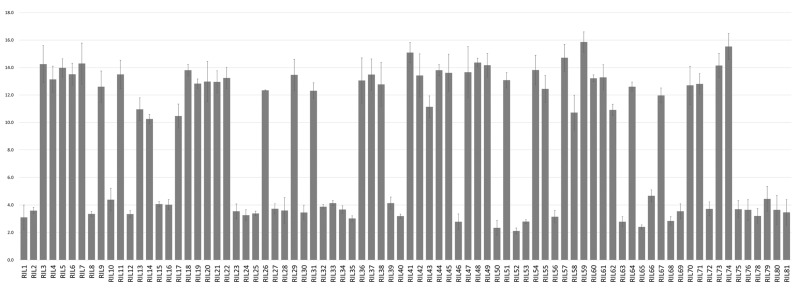
Frequency distribution of average amylose content of each line in maize F_7:8_ RIL population.

**Figure 2 plants-12-00239-f002:**
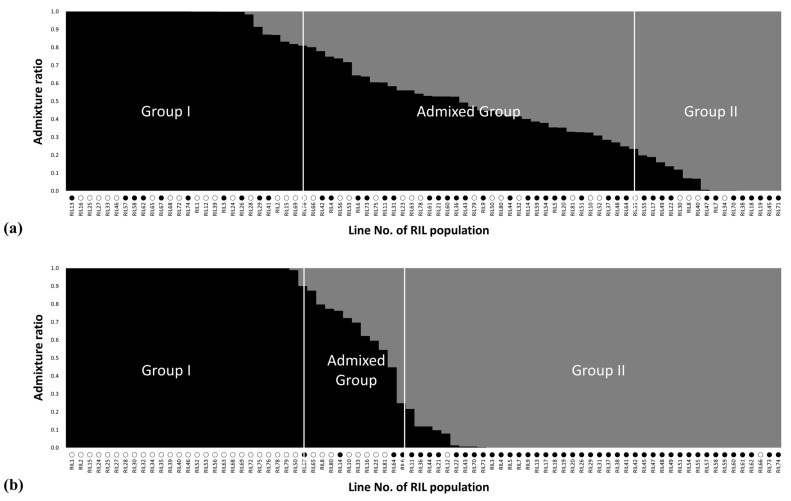
Population structure pattern of 80 RIL population based on SSR and SNP markers for K = 2 and (**a**) 14,968 markers, (**b**) 40 markers by PCA-MLM analysis. ○: low-amylose lines, ●: high-amylose lines.

**Figure 3 plants-12-00239-f003:**
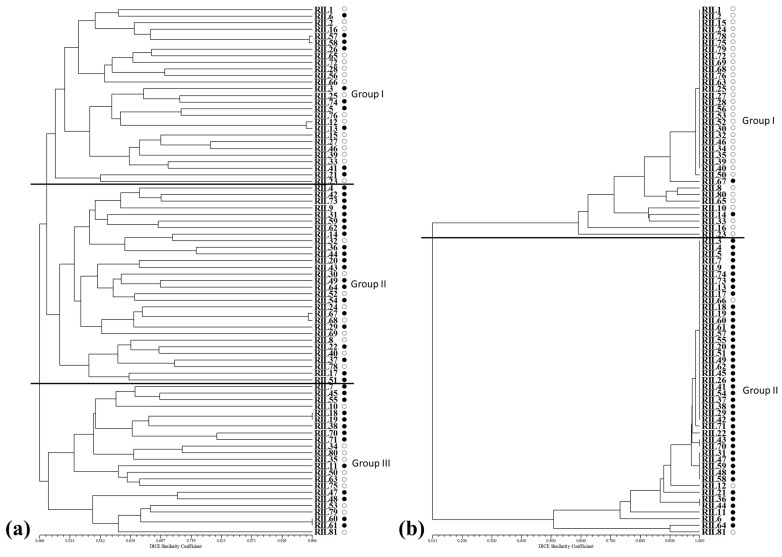
UPGMA dendrogram in maize 80 RIL population and (**a**) 14,968 markers, (**b**) 40 markers by PCA-MLM analysis. ○: low-amylose lines, ●: high-amylose lines.

**Table 1 plants-12-00239-t001:** Results of amylose content evaluation for the two parental lines, all RIL populations, and the low and high groups.

	Female (%)	Male (%)	All (*n* = 80) (%)	Low Group (*n* = 36) (%)	High Group (*n* = 44) (%)
Mean	14.4	2.9	8.8	3.4	13.1
SD	0.6	0.6	0.7	0.5	0.9
Min			2.1	2.1	10.3
Max			15.9	4.7	15.9
*t*-test (*p*)				−45.403 (0.001)	

SD: standard deviation; Min: minimum value; Max: maximum value.

**Table 2 plants-12-00239-t002:** Information of marker–trait association for amylose content using Q + K MLM.

Marker Type	Marker	Chr	Position and Bin Value	*p*-Value	R^2^ (%)	Gene ID	Gene Name	Gene Description	FDR at 0.05	FDR at 0.01	−log10(p)
SSR	umc1634	9	9.03	2.64E − 11	0.870	GRMZM2G121333	-	alpha/beta-Hydrolases superfamily protein	3.34E − 06	6.68E − 07	10.58 **
SNP	PZE-109022525	9	23,019,885	3.40E − 10	0.665	GRMZM2G147319	-	Zinc finger (C3HC4-type RING finger) family protein	6.68E − 06	1.34E − 06	9.47 **
SSR	umc2213	9	9.02	3.66E − 10	0.698	GRMZM5G830776	-	SNARE-interacting protein KEULE	1.00E − 05	2.00E − 06	9.44 **
SNP	SYN10618	9	22,668,399	1.01E − 09	0.620	GRMZM2G092296	rps22a	ribosomal protein S22 homolog	1.34E − 05	2.67E − 06	9.00 **
SNP	SYN34180	9	23,749,286	1.62E − 09	0.601	GRMZM2G118355	-	Histone H3	1.67E − 05	3.34E − 06	8.79 **
SNP	SYN34181	9	23,765,962	1.62E − 09	0.601	GRMZM2G118690	bbr4	BBR/BPC-transcription factor 4	3.01E − 05	6.01E − 06	8.79 **
SNP	SYN34183	9	23,749,944	1.62E − 09	0.601	GRMZM2G118355	-	Histone H3	2.00E − 05	4.01E − 06	8.79 **
SNP	SYN34184	9	23,749,967	1.62E − 09	0.601	GRMZM2G118355	-	Histone H3	2.34E − 05	4.68E − 06	8.79 **
SNP	SYN34187	9	23,750,028	1.62E − 09	0.601	GRMZM2G118355	-	Histone H3	2.67E − 05	5.34E − 06	8.79 **
SNP	PUT-163a-18172151-1376	9	22,689,623	3.50E − 09	0.570	-	-	-	3.67E − 05	7.35E − 06	8.46 **
SNP	SYN10617	9	22,666,672	3.50E − 09	0.570	GRMZM2G092296	rps22a	ribosomal protein S22 homolog	3.34E − 05	6.68E − 06	8.46 **
SNP	PZE-109023492	9	23,538,268	5.80E − 09	0.550	GRMZM2G396553	adc1	arginine decarboxylase1	4.01E − 05	8.02E − 06	8.24 **
SNP	PZE-109023685	9	23,765,279	5.80E − 09	0.550	GRMZM2G118687	TIDP9202	Peptidyl-prolyl cis-trans isomerase (PPIase)	4.34E − 05	8.69E − 06	8.24 **
SNP	PZE-109024175	9	24,345,287	7.42E − 09	0.540	GRMZM2G370155	-	-	4.68E − 05	9.35E − 06	8.13 **
SNP	PZE-109023327	9	23,467,856	8.13E − 09	0.537	GRMZM2G121333	umc1634	alpha/beta-Hydrolases superfamily protein	5.01E − 05	1.00E − 05	8.09 **
SNP	PZE-109023851	9	24,080,158	2.52E − 08	0.493	GRMZM2G082855	er3	erecta-like3	5.34E − 05	1.07E − 05	7.60 **
SNP	PZE-109024053	9	24,147,325	2.52E − 08	0.493	-	-	-	5.68E − 05	1.14E − 05	7.60 **
SNP	wx1	9	9.03	4.30E − 08	0.474	GRMZM2G024993	waxy1	NDP-glucose-starch glucosyltransferase, starch granule-bound	6.01E − 05	1.20E − 05	7.37 **
SNP	PZE-109022101	9	22,340,830	5.91E − 08	0.462	-	-	-	6.35E − 05	1.27E − 05	7.23 **
SSR	umc1893	9	9.02	7.93E − 08	0.473	-	-	-	6.68E − 05	1.34E − 05	7.10 **
SNP	PZE-109021982	9	22,174,224	1.14E − 07	0.509	-	-	-	7.01E − 05	1.40E − 05	6.94 **
SNP	SYN31842	9	22,336,574	1.14E − 07	0.509	GRMZM2G107609	-	Glycine-rich protein	7.35E − 05	1.47E − 05	6.94 **
SNP	PZE-109021558	9	21,857,603	1.22E − 07	0.436	-	-	-	7.68E − 05	1.54E − 05	6.91 **
SNP	PZE-109021565	9	21,884,153	3.57E − 07	0.398	-	-	-	8.02E − 05	1.60E − 05	6.45 **
SNP	PZE-109021581	9	21,886,534	3.57E − 07	0.398	GRMZM2G447455	-	Armadillo repeat-containing kinesin-like protein 3	8.35E − 05	1.67E − 05	6.45 **
SNP	PZE-109022064	9	22,338,567	4.16E − 07	0.459	GRMZM2G107665	-	Aminomethyltransferase	8.69E − 05	1.74E − 05	6.38 **
SNP	PZE-109021851	9	22,066,639	4.32E − 07	0.458	GRMZM2G148106	AY105451	ATP-dependent Clp protease proteolytic subunit	9.35E − 05	1.87E − 05	6.36 **
SNP	SYN9832	9	22,066,507	4.32E − 07	0.458	GRMZM2G148106	AY105451	ATP-dependent Clp protease proteolytic subunit	9.02E − 05	1.80E − 05	6.36 **
SNP	SYN11958	9	25,829,587	7.56E − 07	0.372	-	-	-	9.69E − 05	1.94E − 05	6.12 *
SNP	SYN11959	9	25,830,547	7.56E − 07	0.372	GRMZM2G042080	sod11	superoxide dismutase11	1.07E − 04	2.14E − 05	6.12 *
SNP	SYN11960	9	25,829,682	7.56E − 07	0.372	GRMZM2G042080	sod11	superoxide dismutase11	1.00E − 04	2.00E − 05	6.12 *
SNP	SYN11961	9	25,829,976	7.56E − 07	0.372	GRMZM2G042080	sod11	superoxide dismutase11	1.04E − 04	2.07E − 05	6.12 *
SNP	PZE-109025646	9	25,833,120	2.49E − 06	0.332	GRMZM2G344634	acb2	Acyl-CoA-binding protein2	1.10E − 04	2.20E − 05	5.60 *
SNP	PZE-109026003	9	26,499,170	2.49E − 06	0.332	GRMZM2G393146	-	Putative acyl-activating enzyme 19	1.14E − 04	2.27E − 05	5.60 *
SNP	PZE-109025124	9	25,102,451	5.68E − 06	0.305	GRMZM2G335052	si660005h06c	Putative receptor-like protein kinase family protein	1.17E − 04	2.34E − 05	NS
SNP	SYN34135	9	26,592,678	6.17E − 06	0.302	GRMZM2G136838	krp2	kinesin-related protein2	1.20E − 04	2.41E − 05	NS
SNP	PZE-109020811	9	20,840,270	1.41E − 05	0.276	GRMZM2G134260	hb77	Homeobox-transcription factor 77	1.24E − 04	2.47E − 05	NS
SNP	PZE-109025060	9	25,097,187	1.71E − 05	0.282	-	-	-	1.27E − 04	2.54E − 05	NS
SSR	umc2338	9	9.03	2.75E − 05	0.255	GRMZM2G153792	polm3	polymerase II transcription-mediator3	1.30E − 04	NS	NS
SNP	PZA00583.4	9	20,609,016	3.62E − 05	0.246	GRMZM2G089992	bub3	Budding inhibited by benzimidazoles homolog3	1.40E − 04	NS	NS
SNP	PZE-109020028	9	20,403,620	3.62E − 05	0.246	GRMZM2G000645	-	Coatomer subunit beta’-3	1.37E − 04	NS	NS
SNP	SYN9817	9	20,346,749	3.62E − 05	0.246	GRMZM2G444801	sfp5	sulfate transporter5	1.34E − 04	NS	NS
SNP	PZE-109021152	9	21,138,321	6.06E − 05	0.288	-	-	-	1.44E − 04	NS	NS
SNP	PZE-109044991	9	77,785,458	6.44E − 05	0.229	-	-	-	1.47E − 04	NS	NS
SSR	umc1170	9	9.02	7.77E − 05	0.223	-	-	-	1.50E − 04	NS	NS
SNP	PZE-109021389	9	21,823,525	7.80E − 05	0.275	GRMZM2G144421	saur72	small auxin up RNA72	1.54E − 04	NS	NS
SSR	bnlg244	9	9.02	1.03E − 04	0.240	-	-	-	1.57E − 04	NS	NS
SSR	phi065	9	9.03	1.14E − 04	0.263	GRMZM2G083841	pep1	phosphoenolpyruvate carboxylase1	1.60E − 04	NS	NS
SSR	umc2087	9	9.03	1.18E − 04	0.214	GRMZM2G422069	platz16	PLATZ-transcription factor 16	1.64E − 04	NS	NS
SNP	PZE-109020401	9	20,619,848	1.19E − 04	0.261	-	-	-	1.67E − 04	NS	NS
SSR	umc1700	9	9.03	1.42E − 04	0.206	GRMZM2G422069	platz16	PLATZ-transcription factor 16	1.70E − 04	NS	NS
SNP	PZE-109021109	9	20,982,506	1.54E − 04	0.253	-	-	-	1.74E − 04	NS	NS
SNP	SYN15749	9	20,988,151	1.54E − 04	0.253	GRMZM2G105957	-	-	1.77E − 04	NS	NS

Chr: chromosome; R^2^: phenotypic variation; FDR: false discovery rate; NS: not significant; *, **: Bonferroni thresholds at α = 0.05 and α = 0.01, respectively.

**Table 3 plants-12-00239-t003:** Information of marker–trait association for amylose content using PCA + K MLM.

Marker Type	Marker	Chr	Position and Bin Value	*p*-Value	R^2^ (%)	Gene ID	Gene Name	Gene Description	FDR at 0.05	FDR at 0.01	−log10(p)
SSR	umc1634	9	9.03	1.74E − 10	0.619	GRMZM2G121333	-	alpha/beta-Hydrolases superfamily protein	3.34E − 06	6.68E − 07	9.76 **
SNP	PZE-109022525	9	23,019,885	1.73E − 09	0.48	GRMZM2G147319	-	Zinc finger (C3HC4-type RING finger) family protein	6.68E − 06	1.34E − 06	8.76 **
SSR	umc2213	9	9.02	3.18E − 09	0.481	GRMZM5G830776	-	SNARE-interacting protein KEULE	1.00E − 05	2.00E − 06	8.50 **
SNP	SYN10618	9	22,668,399	4.89E − 09	0.446	GRMZM2G092296	rps22a	ribosomal protein S22 homolog	1.34E − 05	2.67E − 06	8.31 **
SNP	SYN34180	9	23,749,286	1.29E − 08	0.416	GRMZM2G118355	-	Histone H3	1.67E − 05	3.34E − 06	7.89 **
SNP	SYN34181	9	23,765,962	1.29E − 08	0.416	GRMZM2G118690	bbr4	BBR/BPC-transcription factor 4	3.01E − 05	6.01E − 06	7.89 **
SNP	SYN34183	9	23,749,944	1.29E − 08	0.416	GRMZM2G118355	-	Histone H3	2.00E − 05	4.01E − 06	7.89 **
SNP	SYN34184	9	23,749,967	1.29E − 08	0.416	GRMZM2G118355	-	Histone H3	2.34E − 05	4.68E − 06	7.89 **
SNP	SYN34187	9	23,750,028	1.29E − 08	0.416	GRMZM2G118355	-	Histone H3	2.67E − 05	5.34E − 06	7.89 **
SNP	PUT-163a-18172151-1376	9	22,689,623	2.48E − 08	0.396	-	-	-	3.67E − 05	7.35E − 06	7.61 **
SNP	SYN10617	9	22,666,672	2.48E − 08	0.396	GRMZM2G092296	rps22a	ribosomal protein S22 homolog	3.34E − 05	6.68E − 06	7.61 **
SNP	PZE-109024175	9	24,345,287	6.47E − 08	0.367	GRMZM2G370155	-	-	4.01E − 05	8.02E − 06	7.19 **
SNP	PZE-109023492	9	23,538,268	6.86E − 08	0.365	GRMZM2G396553	adc1	arginine decarboxylase1	4.34E − 05	8.69E − 06	7.16 **
SNP	PZE-109023685	9	23,765,279	6.86E − 08	0.365	GRMZM2G118687	TIDP9202	Peptidyl-prolyl cis-trans isomerase (PPIase)	4.68E − 05	9.35E − 06	7.16 **
SNP	PZE-109023327	9	23,467,856	8.06E − 08	0.361	GRMZM2G121333	umc1634	alpha/beta-Hydrolases superfamily protein	5.01E − 05	1.00E − 05	7.09 **
SNP	PZE-109023851	9	24,080,158	3.23E − 07	0.321	GRMZM2G082855	er3	erecta-like3	5.34E − 05	1.07E − 05	6.49**
SNP	PZE-109024053	9	24,147,325	3.23E − 07	0.321	-	-	-	5.68E − 05	1.14E − 05	6.49 **
SNP	PZE-109022101	9	22,340,830	3.59E − 07	0.318	-	-	-	6.01E − 05	1.20E − 05	6.45 **
SNP	PZE-109021558	9	21,857,603	4.12E − 07	0.314	-	-	-	6.35E − 05	1.27E − 05	6.39 **
SSR	umc1893	9	9.02	5.53E − 07	0.318	-	-	-	6.68E − 05	1.34E − 05	6.26 **
SNP	wx1	9	9.03	5.73E − 07	0.305	GRMZM2G024993	waxy1	NDP-glucose-starch glucosyltransferase, starch granule-bound	7.01E − 05	1.40E − 05	6.24 **
SNP	PZE-109021982	9	22,174,224	6.19E − 07	0.356	-	-	-	7.35E − 05	1.47E − 05	6.21 **
SNP	SYN31842	9	22,336,574	6.19E − 07	0.356	GRMZM2G107609	-	Glycine-rich protein	7.68E − 05	1.54E − 05	6.21 **
SNP	PZE-109021565	9	21,884,153	1.64E − 06	0.276	-	-	-	8.02E − 05	1.60E − 05	5.79 *
SNP	PZE-109021581	9	21,886,534	1.64E − 06	0.276	GRMZM2G447455	-	Armadillo repeat-containing kinesin-like protein 3	8.35E − 05	1.67E − 05	5.79 *
SNP	PZE-109021851	9	22,066,639	1.66E − 06	0.326	GRMZM2G148106	AY105451	ATP-dependent Clp protease proteolytic subunit	9.02E − 05	1.80E − 05	5.78 *
SNP	SYN9832	9	22,066,507	1.66E − 06	0.326	GRMZM2G148106	AY105451	ATP-dependent Clp protease proteolytic subunit	8.69E − 05	1.74E − 05	5.78 *
SNP	PZE-109022064	9	22,338,567	2.99E − 06	0.309	GRMZM2G107665	-	Aminomethyltransferase	9.35E − 05	1.87E − 05	5.52 *
SNP	SYN11958	9	25,829,587	1.02E − 05	0.228	-	-	-	9.69E − 05	1.94E − 05	NS
SNP	SYN11959	9	25,830,547	1.02E − 05	0.228	GRMZM2G042080	sod11	superoxide dismutase11	1.07E − 04	2.14E − 05	NS
SNP	SYN11960	9	25,829,682	1.02E − 05	0.228	GRMZM2G042080	sod11	superoxide dismutase11	1.00E − 04	2.00E − 05	NS
SNP	SYN11961	9	25,829,976	1.02E − 05	0.228	GRMZM2G042080	sod11	superoxide dismutase11	1.04E − 04	2.07E − 05	NS
SNP	PZE-109020811	9	20,840,270	2.59E − 05	0.205	GRMZM2G134260	hb77	Homeobox-transcription factor 77	1.10E − 04	NS	NS
SNP	PZE-109025646	9	25,833,120	5.64E − 05	0.186	GRMZM2G344634	acb2	Acyl-CoA-binding protein2	1.14E − 04	NS	NS
SNP	PZE-109026003	9	26,499,170	5.64E − 05	0.186	GRMZM2G393146	-	Putative acyl-activating enzyme 19	1.17E − 04	NS	NS
SNP	PZE-109025124	9	25,102,451	9.59E − 05	0.173	GRMZM2G335052	si660005h06c	Putative receptor-like protein kinase family protein	1.20E − 04	NS	NS
SNP	PZA00583.4	9	20,609,016	9.82E − 05	0.172	GRMZM2G089992	bub3	Budding inhibited by benzimidazoles homolog3	1.30E − 04	NS	NS
SNP	PZE-109020028	9	20,403,620	9.82E − 05	0.172	GRMZM2G000645	-	Coatomer subunit beta’-3	1.27E − 04	NS	NS
SNP	SYN9817	9	20,346,749	9.82E − 05	0.172	GRMZM2G444801	sfp5	sulfate transporter5	1.24E − 04	NS	NS
SNP	SYN34135	9	26,592,678	1.30E − 04	0.166	GRMZM2G136838	krp2	kinesin-related protein2	1.34E − 04	NS	NS

Chr: chromosome; R^2^: phenotypic variation; FDR: false discovery rate; NS: not significant; *, **: Bonferroni thresholds at α = 0.05 and α = 0.01, respectively.

## Data Availability

Data are contained within the article.

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
