# Peer review of "Association Mapping of Amylose Content in Maize RIL Population Using SSR and SNP Markers"

_plants, 2023, doi:10.3390/plants12020239_

Round 1

Reviewer 1 Report

The study focused on amylose content using a maze RIL population derived from dent and waxy inbred lines. The design of the experiment and statistical analysis is correct. English form is good. There are just a few minor suggestions that require consideration.

The Abstract lacks background at the beginning and a conclusion at the end.

Keywords should be arranged alphabetically. Moreover, they should not repeat words from the title.

Please, explain better the novelty of your study.

A scientific hypothesis should be included after the aim of the study.

I do not understand the following sentence: “From one 546 informative markers (541 SSR and 5 SNP) were obtained”.

Axes in Figures should be named.

All abbreviations in Tables should be explained in the table footnote.

In the Materials and methods, software producers should be provided (including name, city, state, and country).

Conclusions should be improved, they cannot be just a synthesis of the results.

In the Reference list, the number of volumes should be written in italics.

After incorporating all the necessary changes, the manuscript can be accepted for publication. 

Author Response

Response to the Reviewer’s comments

Thank you very much for reviewers’ comments on our manuscript. We appreciate their comments and we changed our manuscript according to the reviewer’s suggestions.

Reviewer #1:

The study focused on amylose content using a maze RIL population derived from dent and waxy inbred lines. The design of the experiment and statistical analysis is correct. English form is good. There are just a few minor suggestions that require consideration.

The Abstract lacks background at the beginning and a conclusion at the end.

-> We changed “Abstract” as follow: The ratio of amylose to amylopectin in maize kernel starch is important for the appearance, structure, and quality of food product and processing. This study was to identify quantitative trait loci (QTLs) controlling amylose content in maize through association mapping with simple sequence repeat (SSR) and single nucleotide polymorphism (SNP) markers. Average value of amylose content for an 80 recombinant inbreed line (RIL) population was 8.8 ± 0.7 %, ranging from 2.1 to 15.9 %. We used two different analyses, Q+K and PCA+K mixed linear models (MLMs), and found 38 (35 SNP and 3 SSR) and 32 (29 SNP and 3 SSR) marker-trait associations (MTAs) associated with amylose content. A total of 34 (31 SNP and 3 SSR) and 28 (25 SNP and 3 SSR) MTAs were confirmed in the Q+K and PCA+K MLMs, respectively. This study was detected some candidate genes for amylose content. GRMZM2G118690 encoding BBR/BPC-transcription factor, which is for control of seed development and associated with amylose content of rice. GRMZM5G830776 encoding SNARE-interacting protein (KEULE) and uncharacterized marker, PUT-163a-18172151-1376 were significant with higher R2 value in two difference methods. GRMZM2G092296 were also significantly associated with amylose content in this study. This study focused on amylose content using an RIL population derived from dent and waxy inbred lines using molecular markers. Future studies would be of benefit for investigating the physical linkage between starch synthesis genes using SNP and SSR markers, which would help to build a more detailed genetic map and provide new insights into gene regulation of agriculturally important traits. See Abstract part, in new version manuscript.

Keywords should be arranged alphabetically. Moreover, they should not repeat words from the title.

-> We changed keywords and removed repeat keyword from the title. See Keywords, in new version manuscript.

Please, explain better the novelty of your study.

-> As already explained in the results and discussion section, the purpose of this study was detected genes or genetic regions to control amylose content in maize. Although this study found wx1 on chromosome 9, a well-known gene that directly affects amylose content in maize, we also detected some candidate genes or markers for amylose content, such as GRMZM2G118690, GRMZM5G830776 GRMZM2G092296, Uncharacterized marker, PUT-163a-18172151-1376. Therefore, we suggested the possibility of finding additional genes that regulate amylose content, and we intend to reveal their functions in further studies.

A scientific hypothesis should be included after the aim of the study.

-> We added a scientific hypothesis after aim of this study. See the last part of Introduction, in new version manuscript.

I do not understand the following sentence: “From one 546 informative markers (541 SSR and 5 SNP) were obtained”.

-> This study was used a total of 14,968 SSR and SNP markers. These genotype data were obtained from two separate studies [50, 51]. Thus, 546 genotype data were obtained from study of reference No. 50, and 14,422 SNP genotype data were obtained from another study of reference No. 51. We changed this sentence. See P. 12, L. 2-4 of 4.2 4.2. SSR and SNP genotype and data analysis in RILs, in materials and methods part, in new version manuscript.

Axes in Figures should be named.

-> We added axes name in Figure 1 and 2. See revised Figure, in new version manuscript.

All abbreviations in Tables should be explained in the table footnote.

-> We added footnotes for all abbreviations in Table 1, 2, 3. See revised Tables, in new version manuscript.

In the Materials and methods, software producers should be provided (including name, city, state, and country).

-> Almost of software was cited reference list. We added software producers for some software. See Materials and Methods part, in new version manuscript.

Conclusions should be improved, they cannot be just a synthesis of the results.

-> We improved Conclusion part, and added and removed some sentences. See Conclusion part, in new version manuscript.

In the Reference list, the number of volumes should be written in italics.

-> We changed the volume in the reference list to italics. See References part, in new version manuscript.

Thank you for your review comments.

Reviewer 2 Report

In the submitted manuscript, the authors described the study focused on amylose content using RIL population derived from dent and waxy inbred lines. Future studies would benefit from investigating the physical linkages between starch synthesis genes using SNP and SSR markers, which would help to build a more detailed genetic map and provide new insights into the regulation of agriculturally important trait genes in maize molecular breeding studies.The authors conducted extensively  analyses of the studied markers, combined with the experimental validation.The strong points of the manuscript is a very interesting topic. English style and grammar could be improved

Author Response

Response to the Reviewer’s comments

Thank you very much for reviewers’ comments on our manuscript. We appreciate their comments and we changed our manuscript according to the reviewer’s suggestions.

Reviewer #2:

Dear Editor.

In the submitted manuscript, the authors described the study focused on amylose content using RIL population derived from dent and waxy inbred lines. Future studies would benefit from investigating the physical linkages between starch synthesis genes using SNP and SSR markers, which would help to build a more detailed genetic map and provide new insights into the regulation of agriculturally important trait genes in maize molecular breeding studies. The authors conducted extensively analyses of the studied markers, combined with the experimental validation. The strong points of the manuscript is a very interesting topic. English style and grammar could be improved

-> Thank you very much for your comments on our manuscript. Our manuscript was corrected in English by a native speaker.

Reviewer 3 Report

The authors have made great efforts to identify QTLs controlling amylose content in maize via association mapping with SSRs and SNPs to determine the candidate markers responsible for amylose content for Marker-Assisted-Selection. The attained results are of interest to the readers, agronomists and breeders to better understand about the responsible QTLs for amylose content in maize. Generally, the manuscript has been well written, documented as a scientific paper. However, some comments and suggestions may provide useful information to improve the quality of manuscript as below:

1. Some corrections have been made in the manuscript (see in comments in pdf file)

2. Materials and Methods part, the subsection 4.1. Plant material and amylose content evaluation should be narrated in brief, for example, the parents crossed to develop 80 F 7:8 RILS, did the authors made any selection or analyzing amylose content? 

3. Table 1 should add the unit of the value 

4. The authors should add one separate subsection as "Statistical analysis" for analyzing the data for better understanding

5. All references must be formatted and edited following the submitted journal. Further see comments in pdf file.

Author Response

Response to the Reviewer’s comments

Thank you very much for reviewers’ comments on our manuscript. We appreciate their comments and we changed our manuscript according to the reviewer’s suggestions.

Reviewer #3:

The authors have made great efforts to identify QTLs controlling amylose content in maize via association mapping with SSRs and SNPs to determine the candidate markers responsible for amylose content for Marker-Assisted-Selection. The attained results are of interest to the readers, agronomists and breeders to better understand about the responsible QTLs for amylose content in maize. Generally, the manuscript has been well written, documented as a scientific paper. However, some comments and suggestions may provide useful information to improve the quality of manuscript as below:

  1. Some corrections have been made in the manuscript (see in comments in pdf file)

-> We corrected it based on your comments. See new version manuscript.

  1. Materials and Methods part, the subsection 4.1. Plant material and amylose content evaluation should be narrated in brief, for example, the parents crossed to develop 80 F 7:8 RILS, did the authors made any selection or analyzing amylose content? 

-> As suggested in references No. 49 and 50 in this manuscript, in previous studies by our research team, we constructed RIL population and also performed amylose content analysis for RILs.

  1. Table 1 should add the unit of the value 

-> We added the unit of the value in Table 1. See Table 1, in new version manuscript.

  1. The authors should add one separate subsection as "Statistical analysis" for analyzing the data for better understanding

-> We added one separate subsection as "4.3. Statistical data analysis". See Materials and methods, in new version manuscript.

  1. All references must be formatted and edited following the submitted journal. Further see comments in pdf file.

-> We changed format of references for this journal. See reference list, in new version manuscript.

Thank you for your review comments.

Reviewer 4 Report

Dear authors and the editor,

This study was to identify quantitative trait loci controlling amylose content in maize RIL population, and the result was confirmed in the Q+K and PCA+K MLMs. But in my opinion, there are several problems to be answered and improved.

I. The title seems to be not exactly presentation.

II. In Materials and methods, there lack of the description about QTL mapping in detail, and as well as just two parts which is too simple.

III. Please introduce the methods about the association mapping of amylose content in maize RIL population, and provide relative discussion about the method with the previous reports. In my opinion, there are absolutely deviation separation.

VI. Important findings and highlights should be summarized in the abstract, please consider it.

Author Response

Response to the Reviewer’s comments

Thank you very much for reviewers’ comments on our manuscript. We appreciate their comments and we changed our manuscript according to the reviewer’s suggestions.

Reviewer #3:

This study was to identify quantitative trait loci controlling amylose content in maize RIL population, and the result was confirmed in the Q+K and PCA+K MLMs. But in my opinion, there are several problems to be answered and improved.

  1. The title seems to be not exactly presentation.

-> Thank you very much for your comments, but we think the current title is appropriate.

  1. In Materials and methods, there lack of the description about QTL mapping in detail, and as well as just two parts which is too simple.

-> We appreciate your comments, but we think the explanation of the QTL mapping analysis method in the paper is sufficient. This is because most of the published papers have similar descriptions of QTL mapping analysis methods as we presented in our manuscript.

III. Please introduce the methods about the association mapping of amylose content in maize RIL population, and provide relative discussion about the method with the previous reports. In my opinion, there are absolutely deviation separation.

-> We already described about two methods (QTL mapping using linkage map and association mapping) to detect QTLs in the introduction part. In addition, the statistical methods for association mapping are utilizing GLM and MLM methods using PCA, population, kinship. PCA+K and Q+K MLM methods were used in this study, which were explained in the discussion part. In previous study, it is known that fewer residual false-positive associations occur using PCA model (Yang et al. 2011; doi:10.1007/s11032-010-9500-7). Moreover, the computationally simple PCA model can be substituted for STRUCTURE-generated Bayesian clustering analysis (Q matrix) in the MLM approach (Khazaei et al. 2017; doi:10.3835/plantgenome2017.02.0007). We used Q+K and PCA+K MLMs, and all the SNP and SSR markers detected in PCA+K MLM were overlapped with markers identified by Q+K MLM.

  1. Important findings and highlights should be summarized in the abstract, please consider it.

-> We improved abstract part and added and removed some sentences. See Abstract part, in new version manuscript.

Thank you for your review comments.
